# Aquaponics as a Promising Strategy to Mitigate Impacts of Climate Change on Rainbow Trout Culture

**DOI:** 10.3390/ani12192523

**Published:** 2022-09-21

**Authors:** Christos Vasdravanidis, Maria V. Alvanou, Athanasios Lattos, Dimitrios K. Papadopoulos, Ioanna Chatzigeorgiou, Maria Ravani, Georgios Liantas, Ioannis Georgoulis, Konstantinos Feidantsis, Georgios K. Ntinas, Ioannis A. Giantsis

**Affiliations:** 1Department of Animal Science, Faculty of Agricultural Sciences, University of Western Macedonia, 53100 Florina, Greece; 2Oecon Group, Business & Development Consultants, Frixou 9, 54627 Thessaloniki, Greece; 3Laboratory of Animal Physiology, Department of Zoology, School of Biology, Aristotle University of Thessaloniki, 54124 Thessaloniki, Greece; 4Institute of Plant Breeding and Genetic Resources, ELGO-DIMITRA, 57001 Thessaloniki, Greece

**Keywords:** aquaponics, hydroponics, climate change, sustainability, rainbow trout, intelligent aquaculture, controlled system

## Abstract

**Simple Summary:**

Climate change and overexploitation of natural resources drive the need for innovative food production within a sustainability corridor. Aquaponics, combining the technology of recirculation aquaculture systems (RAS) and hydroponics in a closed-loop network, could contribute to addressing these problems. Aquaponic systems have lower freshwater demands than agriculture, greater land use efficiency, and decreased environmental impact combined with higher fish productivity. Rainbow trout is one of the major freshwater fish cultured worldwide, which, however, has not yet been commercially developed in aquaponics. Nevertheless, research conducted so far indicates that the trout species represents a good candidate for aquaponics.

**Abstract:**

The impact of climate change on both terrestrial and aquatic ecosystems tends to become more progressively pronounced and devastating over the years. The sector of aquaculture is severely affected by natural abiotic factors, on account of climate change, that lead to various undesirable phenomena, including aquatic species mortalities and decreased productivity owing to oxidative and thermal stress of the reared organisms. Novel innovative technologies, such as aquaponics that are based on the co-cultivation of freshwater fish with plants in a sustainable manner under the context of controlled abiotic factors, represent a promising tool for mitigating the effect of climate change on reared fish. The rainbow trout (*Oncorhynchus mykiss*) constitutes one of the major freshwater-reared fish species, contributing to the national economies of numerous countries, and more specifically, to regional development, supporting mountainous areas of low productivity. However, it is highly vulnerable to climate change effects, mainly due to the concrete raceways, in which it is reared, that are constructed on the flow-through of rivers and are, therefore, dependent on water’s physical properties. The current review study evaluates the suitability, progress, and challenges of developing innovative and sustainable aquaponic systems to rear rainbow trout in combination with the cultivation of plants. Although not commercially developed to a great extent yet, research has shown that the rainbow trout is a valuable experimental model for aquaponics that may be also commercially exploited in the future. In particular, abiotic factors required in rainbow trout farming along, with the high protein proportion required in the ratios due to the strict carnivorous feeding behavior, result in high nitrate production that can be utilized by plants as a source of nitrogen in an aquaponic system. Intensive farming of rainbow trout in aquaponic systems can be controlled using digital monitoring of the system parameters, mitigating the obstacles originating from extreme temperature fluctuations.

## 1. Introduction

The ecological crisis and the disruption of the ecological balance of aquatic ecosystems are the main and primary issues negatively influencing both marine and freshwater basins, as well as their inhabitants [1]. They derive from a major impact of climate change, causing alterations in a combination of abiotic factors, such as increased temperatures, rainfall reduction, oxygen availability, and pollution [2]. In this context, temperature, nutrient availability, and oxygen distribution in water levels tend to become a major challenge for global aquaculture [1,3]. Fresh- and seawater temperatures increase due to global warming, and it is reflected in the physiology and welfare of aquatic species through a loss in productivity that occasionally may lead to severe mortalities [3]. Freshwater and wetland ecosystems are, thus, facing extreme impacts owing to the increase in frequency and magnitude of temperature fluctuations, threatening the health of aquatic ectothermic organisms and causing economic losses to aquaculture [4]. In addition, climate change tends to affect human well-being because ecosystem services, such as access to fresh water and the production of aquaculture food, are under threat.

More specifically, the freshwater aquaculture sector is occasionally extremely vulnerable to climate change effects [5,6], which are even more intense in freshwater fish farming, due to concrete raceways built on the flow-through rivers. One of the major freshwater fish reared in this form of aquaculture is the rainbow trout (*Oncorhynchus mykiss*) (Walbaum, 1792), which plays an important role in regional development. Since this fish is mostly reared in raceways [7], it is exposed to a great extent to abiotic factors and is, therefore, expected to be severely affected by climate change [8,9,10].

In recent years, research has been conducted on the development of novel production systems, such as aquaponics, generally characterized as less sensitive to climate change effects. Aquaponic systems combine the co-cultivation of plants, along with freshwater fish farming, in a recycling sustainable system [11,12]. Specifically, the implementation and development of such innovative and sustainable systems are crucial in order to avoid environmental fluctuations that can harm the fish and minimize their production, as well as for the adequate production of aquaculture products with reduced environmental impact.

This study constitutes a literature review of the attempts conducted so far to rear rainbow trout in aquaponic systems, as well as to evaluate the limitations and perspectives. The temperature increase tends to disturb the homeostasis of rainbow trout. Keeping this in mind, the current review further examines the rapid rise in temperature due to climate change and its effect on rainbow trout farming.

## 2. Rainbow Trout Farming, Biological Aspects and Effects of Climate Change

### 2.1. Rainbow Trout Farming Systems and Production Trends

*O. mykiss* is the main freshwater fish species bred in Europe, which possesses significant market value [13]. A large proportion of the rainbow trout farming sector is operated by family businesses throughout Europe, with many of them focusing on primary production, while others hold integrated production facilities performing filleting, smoking, and preparation of various edible products [14,15].

Rainbow trout cultivation comprises mainly grow-out techniques of farming that depend on physical water supplies. For instance, cage farming in water reservoirs (lakes, seawater, and ponds) has gained popularity over the last decade [16]. Although this technique can demonstrate advantages, such as larger cultivating space and bigger production, climate change limits this advantage through acute changes in abiotic factors. Additionally, there are farming units in which water supply is available through springs or rivers, such as flow-through systems in concrete raceways (Figure 1), which are considered to be ideal, exploiting the clarity of the optimal conditions of incoming water and exporting the final wastes back to the riverbanks. Although this technique is common in many countries, many others have established legislation in order to minimize waste exportation to water reservoirs [17]. Additionally, flow-through systems, especially from rivers, have been proven to be dangerous for each other when they co-exist on the same riverbank and the transfer of waste occurs from one to another through the water supply.

However, it should be pointed out that at the highest rate of approximately 70% of the production, farmed rainbow trout is traditionally raised in flow-through raceways built on the route of mountain streams, whereas only approximately 10% of rainbow trout is grown in aquaculture in a water recirculation system (RAS), mainly in countries, such as Denmark [18,19]. They are grown to a size of between 350 and 600 g, which corresponds also to the marketable size, with the following forms of disposal and maintenance: (a) live trout, (b) fresh trout, (c) frozen trout, (d) fresh trout fillets, (e) smoked trout, including fillets, (f) prepared trout [14,15]. Occasionally, the rainbow trout is also bred for its eggs in countries, such as France.

Economic value for rainbow trout starts at 5.6 EUR/kilo in Poland, reaching up to 14.25–22.22 EUR/kg in France, Italy, and Denmark, while even higher market prices of rainbow trout are observed in Austria and Finland [14,15,20]. The main producers of rainbow trout are Iran (164,000 tons), the European Union (183,819 tons), Turkey (103,000 tons), and Peru (55,000 tons). Additionally, a significant amount of rainbow trout is produced in Indonesia, India, Vietnam, Bangladesh, Philippines, Korea, Japan, Egypt, Norway, and Chile. It should be also noted that in some Mediterranean countries, such as Greece and Portugal, despite the numerous existing farming units, the price is relatively low, mainly on account of the consumers’ preference for marine fish. However, in various member states of the European Union (e.g., France, Denmark), rainbow trout is strongly recognized by consumers as a healthy and organic product [21,22]. 

In Greece, approximately 60 units of rainbow trout operate today, with an annual production yield that reaches 1900 tons, worth 6,270,000 million EUR, and four hatcheries [22,23]. The produced fresh rainbow trout is available in two sizes: (a) average weight of 200–500 grams and (b) average weight of 400–500 grams at a price of 2.7 EUR/kg [24].

Interestingly, there has been a sharp decline in the production rates of rainbow trout in recent years in all producing countries (Figure 2), as well as a general fluctuation in the European Union, probably reflecting the physical conditions in combination with the consumers’ annual demand (Figure 3).

It should be also underlined that, in 2018, rainbow trout was the second most produced and marketable species of fish in EU countries, while in 2019, rainbow trout reached the first most produced and marketable species of fish in EU countries, overpassing Atlantic salmon [15,20], emphasizing the high value of the sector for many national economies.

### 2.2. Biological Aspects of Rainbow Trout

*O. mykiss* is native to the shores of the Pacific Ocean of North America and Asia [25,26]. However, it has been introduced and is farmed in at least 82 other countries [27,28,29,30] where it exhibits positive results from an economic and sustainability point of view, as it tolerates a wide range of environmental conditions. This species presents rapid growth rates (1000 grams in 14–16 months), leading to profitable and sustainable farming [31,32].

*O. mykiss* was introduced to Europe in the late 19th century and is now cultivated in almost all European countries [33]. It is usually bred mainly in semi-extensive raceway concrete tanks in most countries inside or outside the EU [25,27,32]. Some rainbow trout populations spend the mature phase of their lives in the ocean, similar to the *Salmo* species [34].

Wild rainbow trout is a psychrophilic species, which usually inhabits the upper reaches of rivers and cold fresh waters [35]. In addition, *O. mykiss* spawns in small to moderately large, well-oxygenated, shallow rivers with gravel bottoms [35]. Rainbow trouts can reach 50 pounds, but farmed trout usually weigh only a few pounds when they are harvested. Rainbow trout is a desirable fish for anglers and consumers, and it is also preferred for farming because of its high-quality flesh, high survival, ability to be fed on pellets, and tolerance to temperatures as high as 21 °C [34]. Additionally, it is capable of being stocked in high raceway/tank densities and possesses great growth rates since it reaches a marketable size in less than a year [36].

Since trouts are carnivorous, their feed should contain high levels of protein, which is a more expensive higher carbon footprint than the pellets appropriate for omnivorous fish [34]. Often, in trout aquaculture, the phenomenon of cannibalism is observed (Figure 4). Cannibalism is an aggressive behavior, which can be caused by stress induced by various population and environmental factors and may occur in two main forms: early larval and late juvenile or adult trout [37]. Farmed rainbow trout requires certain environmental factors (temperature, pH, and dissolved oxygen) for its normal growth and welfare [38,39]. With the intensity of climate change, in recent years, these environmental factors have gradually shifted, affecting the welfare and health of rainbow trout [40,41]. For this reason, researchers chose to use the rainbow trout as an experimental model against climate change based on its capability to tolerate a wide range of environments and handling procedures [40,41].

Moreover, the various anthropogenic interventions in the natural environment, mirrored in the climate change crisis, may have also contributed to the reduction in the existing genetic diversity of the rainbow trout populations. Trout populations are intensively affected by water quality parameters, showing genetic and phenotypic differences when there are changes in water temperature, pH, and oxygen levels [13].

The rapid variation of environmental conditions as a result of climate change may affect the genetic diversity of rainbow trouts with an impact on conservation of the species [42]. Loss of genetic variability influences the long-term survival of populations associated with reduced fitness and an eventual higher risk of extinction. For instance, in a small population of rainbow trout, in successive generations and without gene flow, the probability of mating with closely related individuals is very high, leading to inbreeding and specifically a decrease in the population structure [43,44,45,46,47]. The assessment of reproductive values in all generations showed that multi-leaf selection (with a low rate of inbreeding) has brought an average of 7% genetic gain per generation in the development of rainbow trout of market size [48].

The reduced physical condition of the offspring of closely related individuals is fundamentally associated with the concept of heterozygocity, since the offspring of these species of couples are, necessarily, less heterozygous in their entire genome than extragamous individuals. This lower heterozygocity (i.e., low genetic diversity) makes small populations of rainbow trout more susceptible to low juvenile survival, reduced population growth, small body size, and a decreased number of rainbow trouts reaching adulthood [49]. Genetic diversity plays a crucial role in determining the probability of survival of a population from environmental changes, new pathogens, as well as the average suitability of a population for successive generations [50].

Concerning abiotic factors, water clarity is vital to the efficiency of finding food. Water should not contain harmful solids or harmful gaseous wastes produced during metabolism and respiration. A rainbow trout prefers water temperatures of 13–18 °C [34] and survives in a range of 9–20 °C [19]. Temperature plays a decisive role in a rainbow trout’s growth and metabolic rate, in the duration of egg incubation, in its resistance to various diseases, and in its ability to retain dissolved oxygen from water. Furthermore, water with low temperatures during the summer months (less than 10 °C) should not be used for the cultivation of rainbow trout since its growth rate is significantly slowed down [51].

Furthermore, rainbow trout tolerates adverse pH conditions depending on its life stages. The optimal and acceptable pH ranges of the breeding water also differ. For the development of offspring, the optimal pH range is wide, ranging between 6.5 and 8 [51]. Therefore, it can reach the optimum growth rate in well-aerated, freshwaters with a pH level of 6.7–8.2 [52]. In general, alkaline waters are more suitable for rainbow trout farming in an aquaponic system, being rich in Ca and Mg, elements necessary for the growth of fish bones and the growth of various aquatic organisms offering a better-structured ecosystem [51]. When the pH value of water is more than 9.0 and less than 5.5, then the water is unsuitable for rainbow trout breeding [51].

Water alkalinity also plays an important role in fish production. To ensure rainbow trout’s welfare, alkalinity must be at least 5 ppm. Great growth rates of rainbow trout are usually observed in waters with an alkalinity equal to 150–200 ppm. The lower alkalinity level for trout farming should be 20–50 ppm [51].

The dissolved oxygen (DO) in water ensures the respiration of various aquatic plants and animals. At high water temperatures, the DO content is lower and vice versa. [39]. Specifically, when the temperature rises, the oxygen content of the water generally decreases, while the oxygen consumption of the fish increases. Climate change leads to a reduction in the minimum oxygen consumption (5mg/l) and a decrease in the pH value of water (pH 3) [41,53,54]. This implies that the fish cannot feed effectively and, therefore, extremely high temperatures of the water will block the growth rate of the fish and cause health problems [54].

Apart from the above, optimal and acceptable oxygen concentrations in the water vary depending on the stage of the fish’s development [51]. For the breeding stage, the optimal oxygen content of the breeding water should be close to saturation. More specifically, dissolved oxygen (DO) ranges between 5 and 6 mg/L during egg incubation and in the early stages of pre-larval development [51]. In the adult stages, the acceptable limit of DO content in water may be approximately 4–5 mg/L [55]. During and after feeding, the oxygen consumption of fish increases significantly. During these periods, the demand for oxygen temporarily increases. In short, DO ranges from 5 to 8 ppm and contributes to the growth of plants, fish, and bacteria [55]. Oxygen is among the most important growth regulators in rainbow trout and fish capacity of tanks. The value of the DO in the water for the survival of the rainbow trout shall not be less than 5.5 ppm and >6 mg/L [51]. The lower value of the DO content of water for the survival of the trout is significantly influenced by the dissolved carbon dioxide content of the water. Therefore, at 16.5 °C the rainbow trout can live for 24 h in water with an oxygen content of 2 ppm, while on the contrary, it may not survive an hour or even less if the carbon dioxide content of the water reaches 15 ppm [51]. Higher carbon dioxide content indicates either contamination by organic substances or lack of oxygen [51]. Water hardness for optimal growth of rainbow trout should be between 50 ppm and 200 ppm [51]. Regarding water conductivity, it is well established that it should be between 400 and 450 μS/cm [51]. Water with conductivity values below 100 μS/cm is considered poor for large-scale *O. mykiss* production, while it is optimal for incubation [51].

Usually, wastewater exists in the form of ammonia (${\mathrm{NH}}_{3},$ NH₄⁺) and is released as solid waste (feces). In particular, the amount of unionized ammonia for the survival of the rainbow trout should be <0.001 mg/L, while the number of nitrates should be <0.8 mg/L. [51]. The free form of ammonia is toxic to rainbow trout even at a content of 0.012 ppm because it causes damage to the gills and reduces their growth. Nitrites (${\mathrm{NO}}_{2}^{-}$) are very toxic to rainbow trout when they are at a content of 0.10 ppm. Nitrates (${\mathrm{NO}}_{3}^{-}$) when found in large quantities in the water, create suitable conditions for the growth of aquatic plants. However, the content of nitrates in the rainbow trout farm water should not exceed the value of 100–150 ppm, since above this value, the water becomes toxic [56]. Ideally, in rainbow trout farming, the nitrate concentration should be less than 75 mg/L [56].

### 2.3. Climate Change as a Huge Challenge for Rainbow Trout Aquaculture

Aquaculture is a fast-growing food production sector contributing to global food security. According to FAO, aquaculture food production needs to increase in order to meet future global demand in 2050 [57,58]. Undoubtedly, aquaculture production depends on farming conditions which may be influenced by climate change. The impact of climate change on aquaculture may vary both in type (e.g., water temperature) and extent, depending on climatic zones (temperate, tropical, or Mediterranean), geographical areas (indoor, marine, or coastal aquaculture), types of aquaculture production systems, and aquatic species reared [41]. However, global warming that leads to water temperature increase, will inevitably affect farmed *O. mykiss* species which require lots of oxygen and high-quality water, such as the cold, pure water in the streams where they inhabit and bred in semi-intensive aquaculture systems [41]. 

Climate change contributes to water quality degradation. Water temperature fluctuations can affect the optimal aquatic conditions of rainbow trout farming and increase the costs demands for keeping temperature stable under farming conditions [59]. Extreme weather incidents impose risks in aquaculture production. For example, the increasing frequency and intensity of storms will complicate work, especially, in the cultivation of open water ages and will affect the choice of feeding methods and equipment [54]. Moreover, because nowadays, aquaculture has an increasing and important role in the global production of aquatic animals and food [60,61], aquaculture activities with an increased carbon footprint due to logistics, transport, input power, and feed production contribute to greenhouse gas emissions [54]. The reckless and inefficient use of resources for animal feed, water, and land are deteriorating climate change effects with serious impacts on aquaculture and the bioeconomy [54,59].

Hence, climate change affects and alters environmental factors influencing the welfare of rainbow trout (*O. mykiss*) and, therefore, problems in the growth rate and welfare of the fish may occur [54]. In the waters of the Southern Mediterranean (e.g., shallow, and semi-enclosed bays), during summer, temperatures can exceed 32 °C [62,63]. On the contrary, in the northern part of the Mediterranean, extreme winter temperatures ranging between 10 °C and -20 °C are often observed [64,65,66]. In recent decades, there have been extreme heat waves in Bangladesh due to climate change and global warming [67,68]. In addition, there is an extreme rise in temperature in Vietnam, Australia, and Hawaii [69,70,71]. Moreover, a 2% increase in surface water temperatures in the summer months is expected in Germany, England, and Denmark, and a prolonged increase in temperature by 6% within the period 2040–2059 [9]. During this period, increased summer rainfall is expected in the above countries, increasing the risk of flooding [72]. The impact of the climate is of the utmost importance for the above countries because they constitute the largest part of rainbow trout production and the main export markets. In addition, climate change has a significant impact on rainbow trout aquaculture in north-western European countries. Such effects are warming, deoxygenation, and acidification. Studies have shown that climate change affects non-EU states (e.g., Canada, Turkey, and Japan) by the means of reducing the growth rate of rainbow trout populations [9]. The induction of heat stress of freshwater aquaculture fish (e.g., rainbow trout) in parts of Asia [73], the Mediterranean [74,75], and Australia [76,77,78] during summer (heatwaves) and winter (extreme cold weather events), modifies the energy costs for normal homeostasis, osmotic, and ionic regulation [41,79].

For a homeostasis balance, fish must compensate for a significant amount of energy, which ultimately affects growth, immunity, and resistance to diseases [80,81,82]. Thus, essentially, climate change tends to affect aquaculture-producing rainbow trout in two ways. Firstly, it causes changes in homeostasis [80,81,82] and behavior of both wild and farmed rainbow trout [41,83,84,85]. Secondly, by facilitating the emergence of new pathogens, such as parasites, with adverse effects on fish health.

#### Climate Change and Physiological Homeostasis

Thermal tolerance of rainbow trout (which is sensitive both at extremely high and low temperatures) depends on the genotype, age, stages of development, physical condition, and the history of previous thermal exposure [82,86,87,88,89]. Extreme changes and frequent fluctuations in the temperature of the water affect the endocrine, antioxidant, molecular, immune, and hemato-biochemical functions of rainbow trout [41], thus, having consequences on the growth of fish, reproduction, metabolism, hemato-physiology, and immunology [89,90,91,92] (Figure 5).

Fluctuations outside of the preferable temperature range (7–18 °C) cause a reduction in fish appetite and food utilization [51]. Eventually, fish stop feeding at very low or very high water temperatures. The feeding of rainbow trout intensifies as the water temperature rises. Thus, it will consume food intensively at 18 °C. On the other hand, the digestion process will be decreased at this temperature. Rainbow trout grows better in relation to food consumption when water temperature ranges from 13°C to 15 °C. Optimal, acceptable, and lethal water temperatures also vary depending on the fish’s developmental stage [51]. Therefore, fish mortality is caused at temperatures much lower or higher than ideal [93,94,95].

When fish are exposed to several stress factors, such as climate change effects, levels of catecholamines, corticosteroids, cortisol, glycogen, and lactic acid increase [41]. Thermal stress responses lead to the release of hormones into the blood and tissues, thus, resulting in changes in energy mobilization, ionic balance, immunological responses, cardiac activity, physical performance, growth, and reproduction [96,97,98,99]. Reproductive performance of fish, and in particular, sexual maturity and sex differentiation are affected [100,101,102,103]. Specifically, temperature changes either promote or delay gametogenesis [100,101,102,103]. Extreme high temperatures reduce gene steroidogenesis [101,104,105,106,107], inhibit sperm production, and affect sperm motility in rainbow trout [108]. Moreover, global warming causes changes in the nerve tissues (neuroendocrine function) of the rainbow trout and affects the normal functioning of the central nervous system, which is a major threat to aquaculture production [109,110,111]. Temperature affects gills plasticity through impairments of Na, K, and ATPase activities at 24 °C, therefore, resulting in changes in ionic balance [112,113,114].

In addition, extreme warming contributes to the release of heat shock proteins (HSPs) and activates the genes associated with the immune system [115,116,117,118,119,120,121,122,123,124]. Moreover, the exposure of rainbow trout to extremely high temperatures affects the number of blood cells and cell morphology, causing significant cellular damage and abnormalities [109,110,112]. Therefore, climate change is exposing rainbow trout to thermal stress, affecting fish physiology and behavior [82,88,125]. Moreover, thermal stress affects the redox status of *O. mykiss*. Specifically, temperature-induced oxidative stress leads to significant variability in the antioxidant response of fish, causing cellular and protein damage, which can affect the production yield of rainbow trout [117,126,127]. Chronic thermal and oxidative stress affect the metabolic rate of rainbow trout with extreme aggravation from high respiratory rhythms and oxygen consumption [92,126,128,129,130]. Specifically, a three-fold increase in rainbow trout’s metabolic rate is observed when the water temperature increases by 10 °C [131,132,133]. Under extremely cold conditions, glucose levels in the blood of the fish increase [119,134,135]. More specifically, a sharp decrease in temperature from 11°C to 1 °C during 7 days of extreme cold exposure results in an increase in the blood plasma levels of rainbow trout [136]. Moreover, cold stress decreases enzymatic activity by moderating the lactic acid content and substituting energy [41].

In addition, warming and terrestrial wastewater runoff because of climate change tend to increase the toxicity effects of pollutants on the wildlife of rainbow trout [137]. Additionally, due to global warming, rainbow trout are becoming more sensitive to metal toxicity (e.g., Cu), worsening mitochondrial function (ATP production) [138,139,140,141,142]. Fish exposure to high concentrations of such metals is an important stressor, affecting their biological activities [143,144].

The magnitude of the response of the rainbow trout to stress depends on the rate of temperature variation towards the upper and lower limits and the duration of exposure. Therefore, the increase in the temperature of the water can cause thermal stress on the fish, burdening their immune systems and leading to greater susceptibility to diseases [145,146,147,148,149].

More specifically, continuous temperature fluctuations above 15 °C increase the sensitivity of the rainbow trout to the pathogenic parasite *Tetracapusloides bryosalmonae*, which causes proliferative kidney disease (PKD) [9,150]. In addition, temperature rise contributes to the development of more resistant pathogens, such as the *Rhabdoviridae* virus that causes the disease IHN (infectious hematopoietic necrosis virus) [9] and the parasite *T. Bryosalmonae*. The above diseases cause mass mortality in rainbow trout aquaculture [150].

In short, climate change increases the risk of exposure of *O. mykiss* to infections due to thermal stress. It has been reported that, at 21 °C, pathological changes in liver tissue and inflammation resulting from thermal stress have been observed [151]. Additionally, a significant decrease in immune function in rainbow trout and serious tissue injuries at temperatures above 24 °C was observed [152,153]. In addition, findings support that thermal stress leads to increased levels of cortisol secretion from fish kidneys [41].

## 3. The Potential of Rainbow Trout Farming in Aquaponics

Nowadays, rapidly elevated levels of carbon dioxide (CO₂) have threatened rainbow trout aquaculture and global food security. More specifically, increased CO₂ levels can negatively impact the cardiovascular, metabolic, and homeostatic balance of *O. mykiss* [154]. To increase the global aquaculture production, despite climate change, an expansion of sustainable aquaculture systems (e.g., aquaponic systems, biofloc systems, etc.) is needed in the context of a circular economy. Aquaponics, in combination with selective breeding and thermal acclimation, are promising management strategies that may contribute to the development of a new form of rainbow trout farming [47,48,155,156,157]. Nevertheless, climate change remains a difficult obstacle for examining the genetic diversity of rainbow trout populations, a species sensitive to climate change.

### 3.1. Aquaponics as a Novel Technology

So far, aquaponics research is limited to studies with small populations of fish individuals, mainly focusing on plant features or abiotic and economic parameters. Therefore, it cannot be accurately evaluated whether fish raised in aquaponics systems can overcome the challenges of climate change. However, it can be estimated that various species of fish will be greatly or slightly affected by the effects of climate change in the near future. Specifically, the mainly affected species are freshwater fish of the Salmonidae family, which include rainbow trout (*Oncorhynchus mykiss*) and salmon (*Salmo salar*) [158]. On the other hand, fish of the *Cichlidae* family, such as tilapia (*Oreochromis spp*.), carps, and catfish, are more resistant to climate change [158]. Additionally, tropical ornamental fish, such as *Paracheirodon axelrodi*, are more vulnerable to climate change than temperate fish. Tropical ornamental species have a limited acclimatization capacity, as they thrive in a limited thermal range [158].

Morocco’s inland aquacultures are a prime example of the perception of the effects of climate change in rearing major families with good economic potential [159], such as, for example, some species of trout, i.e., *Salmo akairos*, *Salmo multipunctata, and Salmo pallaryi* [159]. The blue barbell *Pterocapoeta maroccana* (Günther, 1902) is a native species of *Cyprinidae* family, only found in Morocco, included in the IUCN category of endangered species since 2006. *Pterocapoeta maroccana* is the only representative of the genus, threatened by climate change, that species needs to be reared in order to repopulate its habitat, as a freshwater species, with aquaponics and aquaculture potential [159].

As far as salmonoids are concerned, the prevalence of diseases in salmon and trout aquaculture increases as the temperature of the water in cold-water aquaculture rises. Rainbow trout (*Oncorhynchus mykiss*) has been deliberately developed to resist the disease of cold water caused by the bacterium *Flavobacterium psychrophilum*, using genetic improvement [160]. For this reason, it is worth estimating rainbow trout aquaculture in aquaponics systems.

Due to the fact that global food demand will increase by 70–100% by 2050 [161,162] and the key role of the agricultural sector in food security [163], one of the greatest issues of the 21st century is to find a way to produce more food using fewer resources and minimizing environmental impacts [164]. Among the systems currently used by the agricultural sector, aquaculture seems to be the most suitable and convenient for counteracting deficiencies in food production [54]. Modern freshwater aquaculture relies mainly on closed aquaculture systems (RAS), which reuse the same volume of water [165]. In these systems, the rate of water reuse ranges between 80 and 99%, therefore reducing water requirements and the environmental impact of aquaculture [166,167]. The unification of closed aquaculture systems (RAS) and hydroponics—known as aquaponics—improves sustainability and ensures food sufficiency, providing various significant economic and social benefits [52,168]. These innovative sustainable aquaculture systems, already implemented, are characterized as integrated multi-trophic aquaculture (IMTA) or polycultures [169,170].

Specifically, an aquaponic system can be defined as a production system that combines the hydroponic co-cultivation of plants (vegetables, flowers, or herbs) with the rearing of fish in closed greenhouse-type cultivation systems [171]. It is carried out in a closed-circuit aquaculture (recycled farming systems) properly modified, without the addition of chemicals, and requires a minimum volume of water exchange (about 10%) [172]. The addition of water is very low and is only required due to evaporation or operational work [172]. Essentially, the aquaponic system provides by-products, including waste from fish, as inputs (organic fertilizers) to the plants. Aquaponics is excellently applied in collaboration with RAS [14,173], where water is constantly circulating in fish tanks with the help of pipes to the corresponding filters to be filtered (conversion of ammonia and nitrites into nitrates), ending up in plants and then returning cleanly back to fish tanks [174]. Consequently, aquaponic systems maintain better control of natural conditions [175] without requiring large volumes of water, fertile land use [52,176] and increased operating costs due to the non-use of fertilizers [177].

Stocking density and fish species are key factors in aquaponics systems that can significantly affect plant yield and quality, aspects that both farmers and consumers need to be aware of [178]. According to several studies, different plants and fish will have different optimal ratios and largely depend on factors, such as fish stocking density, aquaponics type, plant species, planting density, hydroponic or soil crop production system type, water flow rate, and external factors [179]. Therefore, the density of plants that will be grown in a system of aquaponics depends on the species and the density of the fish [180].

Research by Addler et al. (2000) examined the cultivation of lettuce and basil in hydroponic cultivation in NFT at densities of approximately 6 plants/$m^{2}$ using the waste of rainbow trout (*Oncorhynchus mykiss*). Approximately 109 $m^{3}$ of trout effluent is produced daily. Basil (*Ocimum basilicum* L.) and lettuce (*Lactuca sativa* L.) removed *p* to 0.003 mg/$L^{-1}$ and < 0.001 mg/$L^{-1}$, respectively, from an influent concentration > 0.5 mg/$L^{-1}$ [181].

In addition, research by McMurty et al. (1990) used aquaponic cultivation in sand bed bush beans (Phaseolus vulgaris L. cv. Blue Lake 274), cucumbers (*Cucumis sativus* L. cv Burpee hybrid II), and tomatoes (*Lycopersicon esculentum Mill. Cv. Champion*) in various densities with parallel aquaculture of blue tilapia (*Sarotherodon aureus* L.). Tilapia was bred at a density of 1.68 kg$/m^{3}$, while the crop densities of bush beans were 12,5 plants/$m^{2}$, 16.7 plants/$m^{2},$ and 20 plants/$m^{2}$. For cucumbers, the densities were 1.8 plants/$m^{2}$, 2.6 plants/$m^{2}$, 4 plants/$m^{2},$ and for tomato it was 6.7 plants/$m^{2}$ [182].

Another research by McMurty et al. (1997) was done on floating hydroponic/aquaponic systems with tomato cultivation (*solanum lycopersicum*) and cucumber (*Cucumis sativus*) and parallel aquaculture of two species of tilapia (*Oreochromis mossambicus* and *Oreochromis niloticus*). They tested four different plant densities of 3.83 plants/$m^{2}$, 4 plants/$m^{2}$, 3.94 plants/$m^{2}$, and 4.022 plants/$m^{2}$, based on the biofilter volume tank to fish tank ratio. Plant growth was adequate. Certain nutrients (N, S, K, P, Ca, Mg, and B) fell below sufficiency standards but were above deficiency levels [183].

Aquaponics can be more productive and sustainable under specific circumstances, especially when arable land and water are limited [184]. Thus, aquaponics is more suitable in high altitudes, which are characterized by little water supply and soils with organic matter deficiency. Significant amounts of food can be produced in locations where soil-based agriculture is difficult or impossible [185]. Deserts and arid areas, sandy islands, and urban gardens are the most suitable locations because of reduced water supply [184]. It offers supportive and collaborative methods of producing vegetables and fish and can grow significant amounts of food in locations and situations where soil-based agriculture is difficult or impossible [185].

From an environmental point of view, aquaponics represents a novel technology that improves production efficiency while mitigating environmental impacts (pollution load, including greenhouse gas emissions), diversifying fish production, animal welfare in aquaculture systems, climate change studies, soil depletion, technologies that mitigate the emergence of animal diseases or parasites, and reducing the use of antibiotics, chemical fertilizers, new feed ingredients, and carbon footprint reductions. It prevents the release of aquaculture waste that pollutes water bodies (eutrophication) [169,170,186], allowing greater control of water and production, which makes food safer against possible residues [187]. Therefore, aquaponics is characterized as an alternative method for food production to agricultural practice and sustainability [166]. It applies techniques and procedures aimed at producing biosecurity food, trying to meet the nutritional needs of the population while significantly minimizing the environmental impact by reducing the environmental load [186,187].

Nevertheless, aquaponic systems are still at an early stage of development, and although many new aquaponic companies are starting up in Europe, only a few of them are currently reaching an economically viable minimum production size [188]. The EU countries that have already developed aquaponic farms (commercial or not) are France, Hungary, Belgium, Germany, Ireland, Italy, Slovenia, Spain, the Netherlands, the United Kingdom, the Czech Republic, Denmark, Finland, Romania, and Sweden, with other ones being at the stage of development [188]. There is a marketing opportunity, as consumers are willing to pay more for antibiotic-free products, pesticides, and herbicides and purchase them from local producers. Commercial aquaculture farms could boost their profit by taking advantage of these marketing aspects (quality and local food) [188]. The controlled supply of nutrients, water, and environmental modifications (due to climate change) under greenhouse conditions is one major reason why aquaponics is going to be very successful [189]. In Greece, the technology of aquaponics is at an early stage and is limited to small domestic and research efforts [185,190,191].

The most popular herbs and spices with beneficial properties for culinary purposes that are intended to be grown in aquaponic systems are basil, coriander, chives, parsley, slippery, and mint [192].

However, the most widespread cultivated herbs in aquaponics systems, in which studies are carried out are basil, thyme, coriander, spearmint, and sage [193]. They are the most prevalent in aquaponics systems owing to their fast harvest rate, higher application of these plants in various sectors (cosmetics, pharmaceutical, and food industries), and their significant economic value worldwide [193].

According to several studies, thyme (*Thymus vulgaris*) and basil (*Ocimum basilicum var. Aristotle*) are among the most appreciated aromatic plants by consumers as culinary herbs, but also with multiple uses in the pharmaceutical, cosmetic, or food industries for meat bio-preservation [194,195,196,197].

In the research of Cretu et al. [193], the developmental performance of basil and thyme in hydroponic systems and in an aquaponics system with parallel aquaculture of goldfish (Carassius auratus) were compared. Based on the results, it was evaluated that the production of thyme and basil was more profitable and favorable in the aquaponics system, since at the same time a profitable production of fish was achieved [193].

In another research by Abdel-Rahim et al. [198], who cultivated the aromatic and medicinal plants rosemary (*Rosmarinus officinalis* L.), mint (*Mentha spicata* L.), marjoram (*Origanum majorana* L.), and thyme (*Thymus vulgaris*) with parallel tilapia aquaculture (*Oreochromis niloticus*), remarkable results of plant yield and fish growth were observed [198], with the only drawback being the low consumer demand for tilapia in comparison to other fish species.

Another study is mentioned, where spearmint (*Mentha spicata*) was cultivated in an aquaponic system with parallel aquaculture of African catfish (*Clarias gariepinus*), giving very good yield results of mint, which gave it great commercial importance [199].

To the best of our knowledge, no study has been conducted to report the contribution of rainbow trout to aquaponic cultivation of aromatic and medicinal plants.

### 3.2. Description and Operation of an Aquaponic System

Due to the fact that aquaponics depends on the nutrient cycle [200], the metabolism and excretory products of fish increase water nutrients (excrement and food debris), which are thereafter used by plants at the right proportions for their best possible growth [52,201,202]. Fish wastewater is distributed at plants’ roots, providing the necessary nutrients for plant growth [52]. Then, the plants act as a natural filter, and therefore, the water can be reused by the fish. This creates a small-scale symbiotic ecosystem, where fish and plants can grow [202]. Fish waste (liquids and solids) is filtered with the support of mechanical and biological filters to keep the water clean [52,201,202]. If water stops circulating through the system, the most immediate result will be a reduction in DO oxygen and the accumulation of nutrients in the fish tank [52]. The water in the plant tanks will become anoxic, and the system will collapse [52]. Diseases, mortality, and nutritional deficiencies are the result of an unbalanced system [55]. The proper functioning of an aquaponic system requires the regular control of the system’s balance, through the control of nitrogen levels at the points of entry and exit of water from the hydroponic plant cultivation tank and the aquaculture pool [187,203].

Specifically, the basic function of an aquaponic system is based on the metabolic products of fish as well as the residues of their food, which through the biochemical process of nitrification are oxidized into non-toxic derivatives used/consumed by plants [204,205]. For this reason, aquaponics is also proposed as an alternative way of wastewater management in a closed agricultural fish system with fully controlled conditions [206]. More specifically, ammonia is released from fish as a metabolic product through the gills at a rate of 80% and becomes toxic in high concentrations of fish [164,202,207]. Toxic metabolites (ammonia and nitrite ions) are converted into nitrates, as they are not harmful to fish and are digested at a rate of 97% by plants to meet their requirements for nutrients [55]. Water containing nitrates enters the hydration system and hydroponic growing plants, which are placed on a substrate with water [52,208]. The water is enhanced by nitrates derived from fish waste [185,202,208]. Their roots bind nitrates as well as phosphate ions so that nutrient-free water is reused and is not a restrictive growth factor for fish [206,209]. Thus, the roots of plants are suspended through the substrate in water full of nutrients [210]. Therefore, the water used by a hydroponic system is enriched with nutrients (nitrates, phosphate ions, etc.), and through its constant recirculation, it is absorbed by the roots of plants, acting as a natural fertilizer [188]. According to Lennard (2006), this process consists of a practical technique used to denitrify water in fish tanks in a closed system of recycled fish farming (RAS) [211].

As such, aquaculture waste is diverted through plants and not released into the environment, while at the same time, nutrients for plants are provided in a sustainable, cost-effective, and non-chemical way [168,201,210,212]. In addition, aquaponics has shown that plant and fish production are comparable to those of hydroponics and closed-circuit aquaculture systems [166]. In an aquaponic system, the plant growth is comparable to conventional hydroponics despite the small concentration of nutrients existing in aquaculture water [212,213,214]. Additionally, the production can be higher compared to the production from the soil [215] attributed to increased concentrations of ${\mathrm{CO}}_{2}$ in the air and changes occurring around the roots [213,214,216]. Moreover, the amount of plant biomass produced, in relation to the volume of water used, is 5–10 times greater compared to classical agriculture [217].

The mezzanine method using substrate (MBT—media bed technique) is the most common method applied to small-scale hydration systems [200], while the resources of the filter provide support to plants but also act as a mechanical and biological filter [52]. The mechanical filter is responsible for removing the solid waste of fish (feces) and food remains [52]. It is considered necessary, since the accumulation of waste releases harmful gases, clogs the pipes, changes the flow of water, and therefore is involved in the collapse of the system [52]. On the other hand, the biological filter oxidizes ammonia and nitrites to nitrates through certain *Nitrosomonas* sp. and *Nitrobacter* sp. bacteria [52,207]. These bacteria in an aquaponic system are used to biologically regulate the system, creating the right environment for fish and plants [52]. The system is constantly “fed” ammonia from fish waste that binds to bacteria for the growth and regulation of the biological filter [52,218]. Conversely, the nutrient film technique (NFT) and the deep-water culture (DWC) aquaponic method can be used for commercial aquaponic systems, but both types require a mechanical and a biological filter [200]. The water runoff, assessed by gravity, is concentrated in the filter where the submersible pump is located [200]. Organisms that coexist in an aquaponic system have different requirements, and therefore caution is needed to achieve a balance in the farming system [55]. Fish, plants, and bacteria are characterized by different needs and optimal levels of environmental factors [55,183]. On the other hand, the successful growth of the plant depends on the daily rate of ammonia production and varies depending on the species and nutritional requirements of the plants that will be placed in the cultivation system [186]. The tank should be composed of low-cost materials that ensure the lowest exposure of the fish to potential risks of toxicity (i.e., fiberglass, polyethylene, and stainless steel) [219]. Round tanks with a conical bottom are those recommended due to better water circulation and transportation of solid waste to the center of the tank [52,219].

### 3.3. Perspectives and Requirements of Rainbow Trout Farming in an Aquaponic System

Various fish species exhibit excellent performance, welfare, and growth rates in aquaponic systems [34]. However, the rainbow trout, although it can survive in the cold temperatures required in aquaponic production systems [51], has not been commercially reared so far in such systems. It should be noted that, keeping in mind the abiotic factors generally needed for trout welfare (Table 1), this fish species may represent an optimum candidate for an aquaponic system [34].

Aquaponics can be integrated into raceway rainbow trout cultivation systems, enabling additional revenues (growing vegetables, insects) as well as improvements in wastewater quality [220]. The integration of the two aquaculture systems makes them largely non-consumer and non-polluting users of the water resources [221]. According to already performed experiments, rainbow trout in aquaponics is bred in round plastic or stainless-steel tanks [36]. Additionally, tanks should be covered with a net to prevent fish escape and exogenous factors’ entrance (i.e., leaves) and for protection from various predators [52] (Figure 6).

*O. mykiss* is one of the most well-adapted domesticated fish, cultivated for centuries and the first to be tested in a closed farming system [36,222]. It is a carnivorous fish characterized by high protein demands, consuming approximately 40–50% of their diet. High protein levels release significant amounts of ammonia, nitrates, and biomass [36,217]. Thus, freshwater fish species, such as rainbow trout, are ideal for integrating into a hydroponic crop due to their viability, ease of nutrition, and their effective conversion of food into biomass in combination with their sufficient market value [209].

**Table 1 animals-12-02523-t001:** Requirements of rainbow trout in closed farming systems [34,36,56,209,222,223].

Requirements	Values
Dissolved oxygen (DO)	>5.0–5.5 ppm
Water quality	Good flow, clean freshwater
pH	6.7–8.2
Growth rate	Rapid
Marketable size	350 gr up to one kilo in less than a year
Protein content	High (50%)
Circular tank containing	300 gallons of water
Density	7.26–20 kg/m^3^
Temperature	9–21 °C
Nitrates	<75 mg/L

It is noteworthy to mention that, as the water temperature increases, fish metabolic rates, enzymatic activity, and blood circulation increase, leading to higher oxygen consumption, while at the same time, at the bottom of the tank, more feces and unused food accumulate [55]. Feces and unused food decomposition require oxygen consumption, resulting in a decrease in DO in the water [55]. Remarkably, in cases of high fish stocking density and high temperature of the water, the DO content may significantly reduce, threatening the survival of the rainbow trout [222,224].

Due to its high growth rate and metabolism, rainbow trout has an increased percentage of waste in plants, which leads to a rapid rate of plant growth [223]. The water temperature range for trout aquaculture is optimal for plants that will thrive in cool temperatures, such as leafy greens, e.g., lettuce, beets, carrots, spinach, cabbage, peas, chards, and more [34,210,225]. Secondly, plants with higher nutrient demands, such as tomatoes, cucumbers, peppers, eggplants, broccoli, as well as fruits, such as strawberries and herbs, such as basil, represent also a good option for combined co-cultivation with rainbow trout [52]. For instance, Velichkova et al. (2019) found increased lettuce production (*Lactusa sativa*) in a small-scale rainbow trout NFT and MBT aquaponic system [223].

From an environmental point of view and according to Bordignon et al., (2022), high density stocks of fish tend to determine a lower environmental impact per kg for an increase in fish production, as it regards global warming, cumulative energy demand, and freshwater ecotoxicity [226]. Replacing conventional energy sources with renewable sources, such as solar energy from photovoltaic panels, can significantly reduce the environmental impact resulting from electricity use [226]. Only 3% of the water of traditional raceway aquaculture, which is commonly applied in rainbow trout farming, is used in aquaponics, resulting in significantly reduced demands for water [222]. Thus, according to Addler et al. (2000), the integration of RAS with hydroponic plant cultivation can bring profit by reducing water use and significantly reducing the discharge of unwanted nutrients into the environment [222]. Nevertheless, the effects of water depletion should be carefully assessed from the moment of extraction of the raw materials to the gate of the farm (from production to consumption) [226]. For example, an incubator can significantly increase the water consumption required to produce fish [226].

The balance between the rate of absorption of plants and the amount of food administered to fish in terms of nitrate and potassium budgets is significant [227]. Fish have an upward tendency to retain phosphorus, while plants have different rates of nutrient absorption, more stable in the case of calcium, because they have different periods of evolution in their lifetimes [227]. In addition, conventional treatment alternatives for nitrogen and phosphorus in wastewater, whether using chemical precipitation, physical removals, or land application technologies, represent significant additional costs for the owner of an aquaculture farm [222].

Additionally, for an economically and environmentally sustainable aquaponic system, it is recommended to replace fish feed with a high carbon footprint with alternative feeds while maintaining entirely a green circular economy. Such feeds may be insect meals or extracts from certain plant species. Bordignon et al. (2022) found that the breeding of fish of high commercial value in a system of aquaponics combined with the substitution of their diet with insect meal (*Hermetia illucens*) can contribute to enhancing the competitiveness and attractiveness of aquaponics products, provided that the cost of insect meals is reduced [228]. Moreover, the extract from plants, *Achillea millefolium* and *Acorus calamus,* used as a supplement in the diet of a rainbow trout enhances its growth by improving the food conversion rate (FCR) and its physiological state without affecting the fish flesh quality and productivity of the aquaponic system with water recirculation [227,228].

Based on the research of Petrea S. M. et al. (2013), it was found that the growth of plants in a system of aquaponics is similar to that of conventional, traditional cultivation if appropriate plant densities are used [229]. However, water temperature has shown a significant effect on the growth of both plants and fish in conventional farming systems and in aquaponics. The temperature at 21 °C has been proved to be more suitable for growing in aquaponics, as the growth of fish and plants was higher at this temperature [230]. In contrast, the temperature at 11 °C had a negative effect on the uptake of nutrients by plants, the metabolism of fish, the effectiveness of the biological filter, and the microbial parameters of the system [230]. Based on the same research, it was found that the selection of fish feed and the optimization of nutrients must meet the requirements of plants at low water temperatures in order to produce high-quality plants and fish. With only 39% of the total fixed costs and 63% of the total annual variable costs, the aquaponics system generates 67% of the annual revenue for the combined multi-trophic system [220]. At the same time, revenues from the greenhouse also helped to compensate for the operating losses of the first year of the fish system. The economics of the fish system can be improved by increasing the scale of production [221] and by using fish species with great commercial value and high market prices, e.g., *O. mykiss*. However, the added technical complexity, marketing work, production and market risks, fish and plant diseases, mechanical failures, regulatory changes, and market fluctuations can significantly reduce projected returns [221].

To sum up, the treatment of fishing wastewater with the parallel use of plant hydroponics represents a potentially profitable additional business for fish farmers [221]. Aquaponics is becoming a cost-effective and environmentally friendly system that will encourage classical aquaculture facilities to adapt to the environmental management of their wastewater [231,232]. The breeding of rainbow trout in an aquaponic system with water recirculation is a promising alternative to common flow systems, especially in view of minimizing the depletion of water resources [226].

### 3.4. Comparative Yields of Rainbow Trout with Other Fish Reared in Aquaponics Systems

Among freshwater fish, rainbow trout occupies one of the highest market values, possessing the highest commercial interest in the world. Nevertheless, aquaponics using this fish is performed in a small percentage due to the special climatic conditions it requires for its welfare and proper development.

In 2014, a study in 44 countries around the world found that the most common fish species in aquaponics were tilapia (55%), followed by ornamental fish (koi, goldfish, and tropical fish) (48%) [233]. Similarly, Love et al. reported tilapia (69%), ornamental fish (43%), catfish (25%), and other aquatic animals, including perch (16%), bluegill (15%), trout (10%), and bass (7%) [234]. Villarroel et al. (2016) reported that tilapia (27%), catfish (10%), ornamental fish (8%), trout (7%), bass (4%), and perch (2%) are the most commonly farmed species in Europe [235].

Several ‘easy-to-produce’ fish species, such as blue tilapia (*Oreochromis aureus*), Nile tilapia (*Oreochromis niloticus*), common carp (*Cyprinus carpio*), tench (*Tinca tinca*), and African catfish (*Clarias gariepinus*) have been successfully reared in aquaponic farming [187,236]. Most of these inferences are focused on warm-water fish species, whereas, to our knowledge, no data are available on cold-water species, such as rainbow trout (*Oncorhynchus mykiss*).

One of the most commonly used species in aquaponics is tilapia. This is due to its omnivorous nature, rapid reproduction, and rapid growth [216]. Unlike rainbow trout, this species of fish, together with catfish and cyprinoids, is very resistant and tolerant to a wide range of water parameters, such as the wide temperature range (15–30 °C) and the concentration of free ammonia $\left({\mathrm{NH}}_{3}\right)$, which ranges from 0.2–3.0 mg/L [192].

However, it should not be omitted that the choice of fish species depends upon its economic value, the market demand, and the geographic localization of the production system [210].

Therefore, in the development of aquaponics, the demand of the consumer public plays an important role. Rainbow trout is a globally recognized edible fish in gastronomy, with a high protein content, nutritional value, and good flesh quality compared to tilapia and carp.

Based on the results of Birollo et al. [36], rainbow trout can be successfully grown in a low-tech water-ponic system up to a final density of approximately 17 kg/$m^{3}$ without adverse effects on flesh growth and quality. At the same time, the increased yield and marketability of lettuce is confirmed [36]. More specifically, on average, the carcass yield was 89.0% of the slaughter weight. Meanwhile, the weight and yield of the fillet were 163 g and 49.1% of the slaughter weight, respectively. The marketable lettuce yield was 2.82 ± 0.64 kg/$m^{2}$ at the end of the first cycle and 1.89 ± 0.18 kg/$m^{2}$ at the end of the second cycle [36]. 

In aquaponics, Palm et al. reported a specific growth rate (SGR) of 0.71% $d^{-1}$ for the Nile tilapia (*Oreochromis niloticus*) (174 g initial weight, 5.62 kg/$m^{3}$ initial stocking density) and 0.65% $d^{-1}$ for the African catfish (*Clarias gariepinus*) (initial weight 480 g, initial rearing density of 6.72 kg/$m^{3}$), which are almost similar to those observed in the Maucieri study (0.74% $d^{-1}$ on average in systems, however, both species used in the aforementioned study showed a lower FCR (1.02 on average) compared to the values in the Maucieri tilapia survey (1.71 on average) [237]. Aquaponics at 1.4 kg/$m^{3}$ of baseline stocking density for a 45-d test showed an SGR similar to those observed in the tilapia study [238] (0.82% $d^{-1}$ on average), but dissimilar to those observed in the FCR (2.31 on average) [239].

The mortality rate was very low (3.3% on average), as previously observed for *Cyprinus carpio* bred in the same aquaponic at similar densities (baseline values: 2.5 and 4.6 kg/$m^{3}$, final values: 6.9 and 11.5 kg/$m^{3}$) [238]. To our knowledge, only one study is available on rainbow trout bred in aquaponic systems focused on lettuce micro-biological analysis [240], while commercial aquaponics systems successfully operate in Colombia, Chile, and Canada; other authors found that the mortality rate increased simultaneously when the initial stocking density increased, both in *Cyprinus carpio var. Koi* (from 2.1 to 2.8 kg/$m^{3}$) [241] and in tilapia *Oreochromis niloticus* (from 0.5 to 1.6 kg/$m^{3}$) [242] grown in aquaponics. However, the reaction to different stocking densities can vary dramatically between fish species [243] and is greatly influenced by the biological requirements of fish and water quality.

The carp breeding density influenced the performance of the tested aquaponic system, with better results, in terms of water quality and vegetable production, achieved with an initial rearing density of 2.5 kg/$m^{3}$ [238]. At the end of the trial, fish weighed 446 g on average, reaching a specific growth rate (SGR) of 0.74% $d^{-1}$. The feed conversion ratio for the whole period was 1.71, and the total biomass produced was 5.66 kg/$m^{3}$, on average [238].

Although possible fish-plant combinations are quite high, the main fish and plant crops used are tilapia and herbs. Growers producing more than 1000 kg of fish developed tilapia in addition to other species and always used agglomerated feed. This coincides with the idea of producing fish in greenhouses, and therefore with warmer water [235].

## 4. Conclusions

Climate change may affect the production of the rainbow trout aquaculture sector in the near future, whereas it should be noted that a decline has already been observed in several countries. Climate change highlights the need for innovative food production within a sustainable corridor. Aquaponics, combining the technology of recirculation aquaculture systems (RAS) and hydroponics in a closed-loop network, may contribute to addressing these problems. In particular, rainbow trout farming, usually carried out within flow-through raceways, is vulnerable to increased temperature and oxygen availability. On the other hand, rainbow trout is a product of high quality and market value, characterized by high growth rates, thus, presenting a very promising candidate for rearing in aquaponics. The intensification of aquaponic systems makes it possible to control the production system, fish, and environmental factors using intelligent, innovative technologies. Aquaponics achieves automatic and digital monitoring of aquaculture systems by controlling and mitigating abiotic factors. Although rainbow trout farming has not been commercially developed on a large scale yet, the research conducted so far indicates that it may represent a promising fish candidate with various potential benefits to be reared in aquaponics.

## Figures and Tables

**Figure 1 animals-12-02523-f001:**
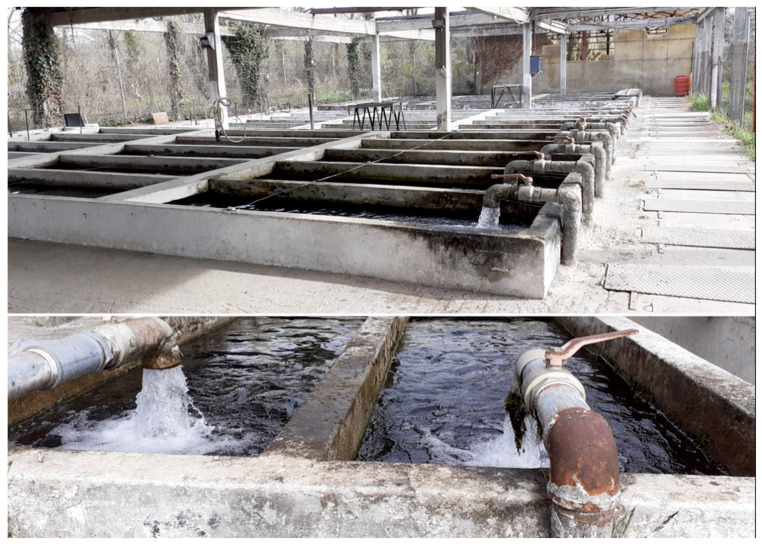
Concrete raceways built on the flow through of a small river in Pella, Greece, utilized in rainbow trout farming.

**Figure 2 animals-12-02523-f002:**
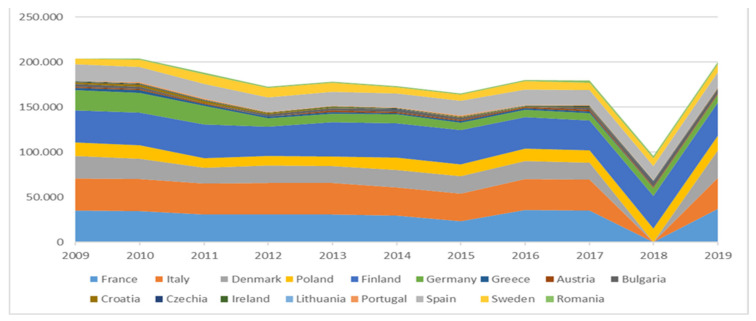
Popular countries with significant production of rainbow trout (tons) in the years 2009–2019 (EUMOFA 2020–2021; EUROSTAT; FEAP 2014–2019).

**Figure 3 animals-12-02523-f003:**
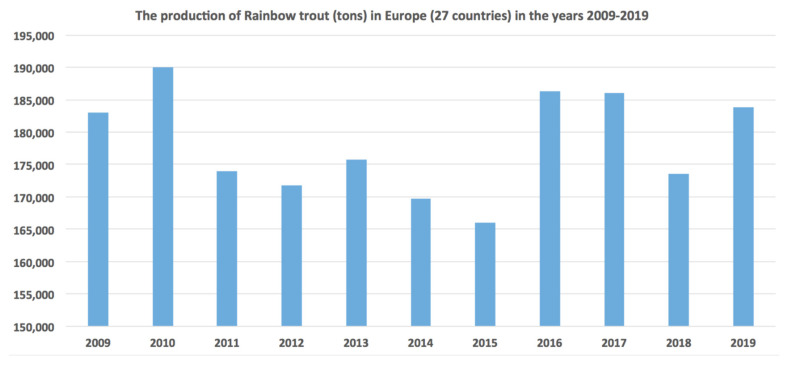
The production of rainbow trout (tons) in Europe (27 countries) in the years 2009–2019.

**Figure 4 animals-12-02523-f004:**
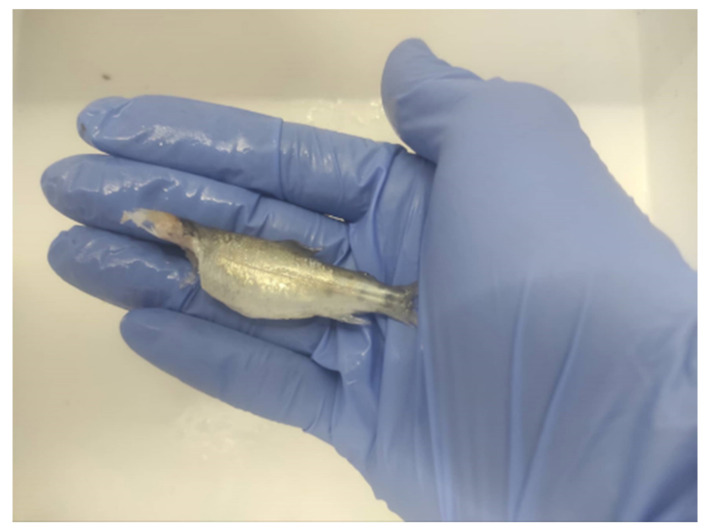
Rainbow trout juvenile after cannibalism observed in an experimental farm.

**Figure 5 animals-12-02523-f005:**
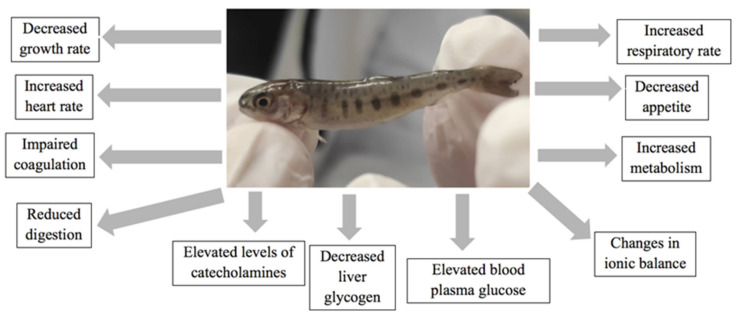
Schematic representation of the major climate change stressors in rainbow trout farming.

**Figure 6 animals-12-02523-f006:**
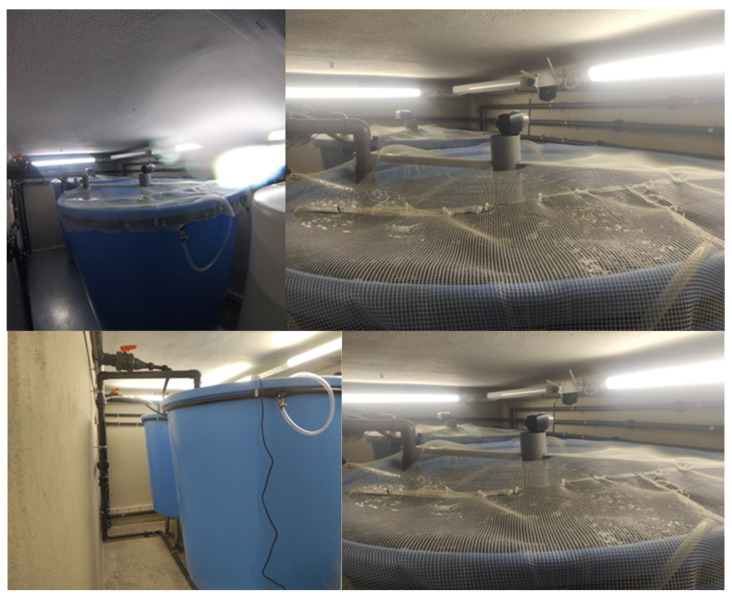
Round plastic tanks covered by a fish protection grid in an experimental research of rainbow trout aquaculture in an aquaponic system in Greece.

## Data Availability

Not applicable.

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
