# Peer review of "Aquaponics as a Promising Strategy to Mitigate Impacts of Climate Change on Rainbow Trout Culture"

_animals, 2022, doi:10.3390/ani12192523_

Round 1

Reviewer 1 Report

The manuscript entitled "Progress and challenges on rearing rainbow trout in aquaponic systems: A promising perspective in the frame of diminishing climate change effects on rainbow trout aquaculture” by Vasdravanidis et al. aims to argue the necessity of breeding rainbow trout in aquaponic systems and its challenging/perspectives by means of climate change.

The manuscript discusses a lot about rainbow trout biology and ecological factors and how these factors are affected by climate change with an adequate reference list. They discuss also, about potential behavior (e.g. cannibalism) and physiological processes (e.g. homeostasis) driven by climate change. I’d recommend the authors to also include studies (if any) that associate fitness-related traits and a given genetic profile of the species in a population level. The authors should consider the level of inbreeding possible found in a farm and how this could affect the behavior or physiological processes in conjunction with climate change. I believe this will enhance their arguments.

Since this is a review paper on aquaponics and climate change, I’d expect to see how other aquaponic systems in other species, or even in the same species, have overcome or not the climate change challenges. We need to see such comparisons to re-evaluate our goals and perspectives.

Overall, it is a well-written manuscript and I suggest the authors consider the above suggestions/recommendations before publication. Therefore I suggest  to revise the manuscript according to the above suggestions.

Author Response

The manuscript entitled "Progress and challenges on rearing rainbow trout in aquaponic systems: A promising perspective in the frame of diminishing climate change effects on rainbow trout aquaculture” by Vasdravanidis et al. aims to argue the necessity of breeding rainbow trout in aquaponic systems and its challenging/perspectives by means of climate change.

The manuscript discusses a lot about rainbow trout biology and ecological factors and how these factors are affected by climate change with an adequate reference list. They discuss also, about potential behavior (e.g. cannibalism) and physiological processes (e.g. homeostasis) driven by climate change. I’d recommend the authors to also include studies (if any) that associate fitness-related traits and a given genetic profile of the species in a population level. The authors should consider the level of inbreeding possible found in a farm and how this could affect the behavior or physiological processes in conjunction with climate change. I believe this will enhance their arguments

Response: We are grateful for the suggestions of the reviewer and the recognition of the extensive reference list. In accordance to this suggestion, a small part was added in the section “2.2. Biological aspects of Rainbow trout”, discussing the fitness of rainbow trout in relation with genetic profile.Also, previous studies are presented, suggesting the effect of inbreeding on the fitness of rainbow trout as a consequence of climate change.

Since this is a review paper on aquaponics and climate change, I’d expect to see how other aquaponic systems in other species, or even in the same species, have overcome or not the climate change challenges. We need to see such comparisons to re-evaluate our goals and perspectives

Re: According to the first reviewer's comment, several examples of fish bred experimentally in aquaponics systems were added. However, it has not yet been precisely assessed whether fish in aquaponics systems have overcome the challenges of climate change, and this fact was mentioned in the revised manuscript (please see section 3.1 in the revised manuscript).

Overall, it is a well-written manuscript and I suggest the authors consider the above suggestions/recommendations before publication. Therefore, I suggest to revise the manuscript according to the above suggestions.

Re: Following and having addressed and embedded all suggestions, comments, and recommendations of the first reviewer, we hope and believe that the quality of presentation of the revised manuscript has been deeply improved.

Reviewer 2 Report

vc

·         The present Ms is a direct response to the mitigation and resilient strategies to climate change impacts on aquaculture. Aquaponics are a new cutting-edge intervention to improve the sustainability of fish production while conservation the environment and creating wealth. The review is worthwhile to inform and educate the world populace.

·         I have the following comments

·         The title is too long and vague. It is ambiguous to the reader what the author is writing in to documents as there are many terminologies explaining several and different things. the second part is simply misleading. Find a way of having a catchy, simple, short topic on: aquaponics as a strategy to mitigate impacts of climate change on trout culture

·         In the summary (line 26-27) final sentence please that trout not this fish.

·         Figure 3 please improve the graphics of the diagram.

·         Introduction:  Line 85-87 please use simple English and avoid colloquial words.

·         Please include a section to compare trout with other species of fish in perfomance in aquaponics and draw recommendations 

·         Please indicate plant densities required in aquaponics

·         Also recommend highly beneficial plants to be used in aquaponics systems , profile of nutritional and benefits of such plants are important .

Author Response

  • The present Ms is a direct response to the mitigation and resilient strategies to climate

change impacts on aquaculture. Aquaponics are a new cutting-edge intervention to improve the sustainability of fish production while conservation the environment and creating wealth. The review is worthwhile to inform and educate the world populace.

Response: We feel pleased that the reviewer recognized and agreed with the main idea of our work, as well as for finding it important.

  • I have the following comments
  • The title is too long and vague. It is ambiguous to the reader what the author is writing in to documents as there are many terminologies explaining several and different things. the second part is simply misleading. Find a way of having a catchy, simple, short topic on: aquaponics as a strategy to mitigate impacts of climate change on trout culture

Re: In accordance to the reviewer’s comment, the title was shortened and modified taking into consideration her/his suggestion.

  • In the summary (line 26-27) final sentence please that trout not this fish.:

Re: Corrected following the reviewer’s comment.

Figure 3 please improve the graphics of the diagram.

Re: The diagram was replaced by a better one in terms of graphics

Introduction: Line 85-87 please use simple English and avoid colloquial words

Re: This part was rephrased in an effort to be more easy to understand, as suggested by the second reviewer.

Please include a section to compare trout with other species of fish in perfomance in aquaponics and draw recommendations

Re: Following the suggestion of the reviewer,a new section (3.4. Comparative yields of rainbow trout with other fish reared in aquaponics systems) was included indicating the yields of rainbow trout and some other fish in aquaponicsystems.

Please indicate plant densities required in aquaponics

Re: Following the suggestion of the second reviewer, a paragraph has been added listing the most common plant species grown in aquaponics systems and their appropriate densities (please see section 3.1).

Recommend highly beneficial plants to be used in aquaponics systems, profile of nutritional and benefits of such plants are important.

Re: The necessary information in the revised manuscript was added in detail as recommended by the second reviewer in the end of section 3.1.

Reviewer 3 Report

The general structure of the work is simply and well described, and the text is clear and concise.

In my opinion, some revisions are needed to make this article acceptable for publication:

-Line 239: NH4+ 

Author Response

Some revisions are needed to make this article acceptable for publication

Re: We are grateful to the third reviewer for recognizing the interest of our work. Following and having addressed and embedded all suggestions, comments and recommendations of the reviewers, we hope and believe that the quality of presentation of the revised manuscript has been deeply improved in terms of explaining the details of the work and clarity.

Line 239: NH₄⁺

Re: Corrected according to the third reviewer’s comment.